# Phenotypic Plasticity Index as a Strategy for Selecting Water-Stress-Adapted Coffee Genotypes

**DOI:** 10.3390/plants12234029

**Published:** 2023-11-30

**Authors:** Cyntia Stephania dos Santos, Ana Flavia de Freitas, Glauber Henrique Barbosa da Silva, João Paulo Pennacchi, Milene Alves Figueiredo de Carvalho, Meline de Oliveira Santos, Tatiana Silveira Junqueira de Moraes, Juliana Costa de Rezende Abrahão, Antonio Alves Pereira, Gladyston Rodrigues Carvalho, Cesar Elias Botelho, Vania Aparecida Silva

**Affiliations:** 1Southern Minas Regional Unit, Agricultural Research Corporation of Minas Gerais (Epamig), Lavras 37200-900, MG, Brazil; cyntia.s.santos@hotmail.com (C.S.d.S.); melineoli@hotmail.com (M.d.O.S.); tatianajunqueira2@gmail.com (T.S.J.d.M.); julianacosta@epamig.br (J.C.d.R.A.); tonico.epamig@gmail.com (A.A.P.); cesarbotelho@epamig.br (C.E.B.); 2Pitangui Institute of Agricultural Technology (Itap), Pitangui 36650-000, MG, Brazil; ana.freitas@epamig.br; 3Department of Food Science, Universidade Federal de Lavras, Lavras 37200-900, MG, Brazil; glaubermav@hotmail.com; 4Department of Biology, Universidade Federal de Lavras, Lavras 37200-900, MG, Brazil; jppennacchi@gmail.com; 5Brazilian Agricultural Research Corporation (Embrapa Cafe), Brasilia 70770-901, DF, Brazil; milene.carvalho532@gmail.com

**Keywords:** *Coffea arabica* L., phenotypic plasticity, water stress, genotypic selection, adaptive strategy, physiological breeding

## Abstract

The adaptive potential of plants is commonly used as an indicator of genotypes with higher breeding program potential. However, the complexity and interaction of plant metabolic parameters pose a challenge to selection strategies. In this context, this study aimed to explore phenotypic plasticity within the germplasm of Hybrid Timor coffee. Additionally, we assessed the utility of the multivariate phenotypic plasticity index (MVPi) as a promising tool to predict genotype performance across diverse climatic conditions. To achieve this, we evaluated the performance of seven accessions from the Hybrid Timor germplasm in comparison to the Rubi and IPR 100 cultivars, known for their susceptibility and resistance to drought, respectively. The experiment took place in a greenhouse under two conditions: one with normal soil moisture levels near maximum capacity, and the other with a water deficit scenario involving a period of no irrigation followed by rehydration. Data on physiological and biochemical factors were collected at three stages: before applying the water deficit, during its imposition, and after rehydration. Growth data were obtained by the difference between the beginning and end of the experimental period Furthermore, field evaluations of the productivity of the same genotypes were carried out over two consecutive seasons. Based on physiological and biochemical assessments, the MVPi was computed, employing Euclidean distance between principal component multivariate analysis scores. Subsequently, this index was correlated with growth and productivity data through linear regressions. Our findings reveal that the plastic genotypes that are capable of significantly altering physiological and biochemical parameters in response to environmental stimuli exhibited reduced biomass loss in both aerial and root parts. As a result, this positively influenced their productivity. Enhanced plasticity was particularly prominent in accessions from the MG Germplasm Collection: MG 311—Hybrid Timor UFV 428-02, MG 270—Hybrid Timor UFV 377-21, and MG 279—Hybrid Timor UFV 376-31, alongside the Rubi MG 1192 cultivar. The MVPi emerged as a valuable instrument to assess genotype adaptability and predict their performance under varying climatic scenarios.

## 1. Introduction

The cultivation of coffee holds significant socio-economic importance in Brazil, but its productivity is facing challenges due to adverse climatic conditions like increasing temperatures and unpredictable rainfall patterns [1,2]. Severe drought periods can lead to plant mortality, and even moderate droughts can have detrimental effects, impacting coffee production. However, plants exhibit the capacity to adapt to adverse conditions, including water deficits, by activating physiological, biochemical, and morphological mechanisms to sustain vital functions [3]. Phenotypic plasticity is a crucial mechanism by which organisms, including plants, adapt to environmental changes, contributing to water stress tolerance. It enables plants to adapt to diverse environmental conditions such as water availability, nutrients, light, and temperature, proving valuable in predicting plant responses to climate change [4].

In drought conditions, as soil moisture gradually decreases, the water potential in the xylem also drops, increasing the risk of hydraulic failure. In response to water restriction, plants initiate a physiological response by closing their stomata [5]. This serves to mitigate excessive transpiration and conserve water [6]. However, this closure of the stomata can lead to a reduction in the photosynthetic rate, as it restricts the intake of CO_2_. Additionally, it affects the plant’s water potential and solute transportation [7,8]. This imbalance between light absorption and utilization can result in an overproduction of reactive oxygen species (ROS), potentially causing cellular damage and, in severe cases, leading to plant mortality [9,10]. Limited water availability not only leads to a loss of turgor, impacting cell division and extension, but also results in reduced overall plant growth [11,12]. It achieves this by reducing exposed surface area, lowering water demand, and minimizing excessive transpiration [11].

These responses to water stress may vary among species and genotypes of the same species, implying varying aptitudes for dealing with this condition, i.e., they exhibit higher levels of plasticity [13]. Considering that climate changes significantly impact global climate, comprehending coffee’s phenotypic plasticity in response to water stress and identifying genotypes with heightened adaptative capacity assumes paramount importance. This aids in developing management and genetic improvement strategies to maximize productivity and sustainability of coffee cultivation under water restriction [14].

In breeding programs, the evaluation and selection of genotypes traditionally involve a meticulous analysis of their phenotypic responses to specific parameters. This process aims to pinpoint genotypes demonstrating a notable capacity to influence variations in the target variable, often linked to productivity. However, the surge in genotypes for evaluation, combined with advanced high-throughput phenotyping methods, has led to more intricate and potentially less efficient processes [15]. Drought tolerance, a complex trait influenced by multiple genes, necessitates the assessment of various morphological, physiological, and molecular characteristics to identify tolerant plants [1,16,17,18]. This evaluation often spans multiple coffee crop cycles due to its perennial nature.

In this context, the possibility of employing a multivariate phenotypic plasticity index emerges as an alternative for genotype evaluation in genetic improvement programs. This index not only considers the isolated phenotypic responses of each parameter but also how these responses harmonize under diverse environmental conditions. The multivariate phenotypic plasticity index (MVPi) proposed by Pennacchi et al. [19] consolidates multiple phenotypic characteristics into a single plasticity marker based on parameter variance and Euclidean distances in a multidimensional space, enhancing efficiency in genotype evaluation.

Building upon our earlier work, when our research group identified Hybrid Timor accessions from the Epamig Germplasm Collection that displayed varying responses to water stress [17], our recent investigation aimed to delve deeper. This time, we focused on exploring the phenotypic plasticity present within the germplasm of Hybrid Timor coffee. Additionally, we assessed the utility of the MVPi as a promising tool to predict genotype performance across diverse climatic conditions. Based on the analysis of phenotypic plasticity in Hybrid Timor coffee genotypes subjected to water stress, we hypothesize that significant variations occur in adaptive responses among different accessions. Furthermore, we posit that the use of MVPi may lead to a more efficient assessment of genotype performance under diverse climatic conditions. This ultimately contributes to the development of coffee varieties better suited to limited water availability. The potential to develop coffee plants adapted to low water availability renders this investigation highly pertinent.

## 2. Materials and Methods

Seven accessions of *Coffea arabica* L. from the Germplasm Collection (GC-MG) of the Minas Gerais Agricultural Research Agency (Empresa de Pesquisa Agropecuária de Minas Gerais—EPAMIG, Patrocínio, Brazil), chosen based on productivity parameters, beverage quality, and disease resistance (Table 1), along with two control cultivars, the drought-tolerant (IPR 100) [17] and drought-sensitive (Rubi MG 1192) [20], were subjected to physiological, biochemical, and growth evaluations under controlled greenhouse conditions. Additionally, field productivity evaluations were conducted.

### 2.1. Greenhouse Experiment

For seedling formation, seeds of selected *Coffea arabica* L. accessions were germinated in sand until reaching the stage of cotyledon leaf emergence. Subsequently, they were transplanted into 120 mL polyethylene tubes containing a substrate for plants based on pine bark, peat, and expanded vermiculite enriched with macro- and micronutrients from the brand Tropstrato HT. The seedlings were then kept in a nursery until they developed four pairs of true leaves and were acclimated to 100% solar luminosity for a period of 30 days.

Following this duration, they were transplanted into 20 L polyethylene pots filled with a substrate mixture comprising 3 parts topsoil, 1 part sand, and 1 part bovine manure (3:1:1 ratio). The plants were shaded at 50% luminosity retention using a screen. They were cultivated in a greenhouse for eleven months until they attained sufficient growth and leaf area for the imposition of water treatments and subsequent leaf sample collection. Fertilization was administered based on substrate analysis and regional crop recommendations. Air temperature and relative humidity were monitored using an ACU-RITE brand device.

The plants were irrigated to maintain the soil at 100% available water for eleven months, with the hydraulic treatment initiated in April 2019. In the first hydraulic treatment, the plants were maintained with the soil at 100% available water from April 2019 until the end of the experimental period (irrigated—I). In the second treatment, irrigation was completely suspended (non-irrigated—NI) until the majority of non-irrigated plants reached a predawn water potential of −3 MPa [21]. Once this water potential was reached, which occurred 33 days after the stress imposition, irrigation was resumed, again maintaining the soil at 100% available water.

For the assessments of gas exchange and biochemical analyses, three distinct periods were considered: 1—prior to the imposition of water deficit; 2—at 25 days after the imposition of water deficit (when the majority of plants reached a predawn water potential around −2 MPa); and 3—at 17 days after the resumption of irrigation. For the assessments, one plant per experimental plot was employed.

Gas exchange assessments were conducted between 8 and 11 a.m. under artificial light (1000 µmol m^−2^ s^−1^) using a portable infrared gas analyzer (IRGA LICOR—6400 XT, LI-COR Bioscienses, Lincoln, NE, USA). Measurements included net photosynthetic rate (A—µmol CO_2_ m^−2^ s^−1^), stomatal conductance (gs—mol H_2_O m^−1^ s^−1^), transpiration rate (E—mmol H_2_O m^−2^ s^−1^), and instantaneous water use efficiency (WUE—µmol CO_2_/mmol H_2_O m^−2^ s^−1^) (A/E). Water potential was determined using a pressure chamber (PMS Instruments Plant Moisture—Model 1000, PMS Instrument Company, Albany, NY, USA) before sunrise.

Regarding biochemical analyses, a fully expanded leaf was collected from each plant in the afternoon (between 12 and 1 p.m.), promptly submerged in liquid nitrogen, and preserved in an ultra-freezer (−80 °C). Subsequent maceration and extraction were performed to quantify hydrogen peroxide (H_2_O_2_), lipid peroxidation, antioxidant metabolism, and ascorbate content.

For H_2_O_2_ quantification, 100 mg of plant material was macerated in liquid nitrogen and polyvinylpolypyrrolidone (PVPP), followed by homogenization in 0.1% (*w*/*v*) trichloroacetic acid (TCA). The samples were centrifuged at 12,000× *g* for 15 min at 4 °C. The supernatant was mixed with a reaction solution containing 10 mM potassium phosphate buffer (pH 7.0) and 1 M potassium iodide. The H_2_O_2_ concentration was determined by measuring absorbance at 390 nm using a standard curve of known H_2_O_2_ concentrations [22], with modifications. Lipid peroxidation was assessed by quantifying thiobarbituric acid reactive species (TBA), as described by Buege and Aust [23]. The aliquots were added to the reaction medium composed of 0.5% (*w*/*v*) thiobarbituric acid (TBA) and 10% (*w*/*v*) trichloroacetic acid (TCA). Subsequently, the medium was incubated at 95 °C for 30 min. The reaction was halted by rapid cooling on ice, and readings were taken at 535 nm and 600 nm. TBA forms reddish-colored complexes, such as malondialdehyde (MDA), a secondary product of the peroxidation process. The concentration of MDA was calculated using the following equation: [MDA] = (A535 − A600)/(ξ × b), where ξ (molar extinction coefficient = 1.56 × 10^−5^); b (optical path = 1). Peroxidation was expressed in mmol of MDA.g^−1^ (FW—fresh weight).

For the determination of antioxidant enzyme activity, 100 mg of plant material was macerated in liquid nitrogen and polyvinylpolypyrrolidone (PVPP), followed by homogenization with 3.5 mL of the following extraction buffer: 100 mM potassium phosphate (pH 7.8), 0.1 mM EDTA, and 10 mM ascorbic acid. The extract was centrifuged at 13,000× *g* for 10 min at 4 °C. The supernatants were collected and used for the analysis of the enzymes catalase (CAT), superoxide dismutase (SOD), and ascorbate peroxidase (APX) [24].

For the determination of catalase (CAT) activity, the protocol of Mengutay et al. [25] with modifications was employed. Aliquots of the samples were added to the incubation medium composed of phosphate buffer (45 mM, pH 7.6), Na_2_EDTA (0.1 mM) (dissolved in the phosphate buffer) and hydrogen peroxide (10 mM). Enzyme activity was determined by the decrease in absorbance at 240 nm every 15 s for 3 min, monitored by the consumption of hydrogen peroxide. The molar extinction coefficient used was 36 mM^−1^cm^−1^.

The ascorbate peroxidase (APX) activity was determined by monitoring the oxidation rate of ascorbate at 290 nm over 3 min, following Nakano and Asada’s methodology (1981), with modifications. Ascorbate quantification was performed using 50 mg of plant material, which was macerated in liquid nitrogen and PVPP and then homogenized with 5% trichloroacetic acid (*m*/*v*). After centrifugation, the samples were added to a reaction mixture containing 5% trichloroacetic acid, 99.8% ethanol, ascorbic acid, phosphoric acid (0.4% in ethanol), bathophenanthroline (0.5% in ethanol), and ferric chloride (III) (0.03% in ethanol). The mixture was incubated, and readings were taken at 534 nm using a standard curve with known concentrations [26].

All biochemical analyses were conducted using 96-well microtitration plates, and readings were performed using a Synergy TM HTX multimode microplate reader.

At the end of the experimental period, dry weights of the aboveground plant parts (MSPA—g), root dry weight (MSR—g), and total dry weight (MST—g) were obtained. Roots were washed, and aboveground parts were separated from the root system. The samples were then dried in a forced-air oven at 70 °C for 72 h, and the dry weight was measured using a precision balance.

### 2.2. Field Experiment

The GC-MG was established in 2005 at the EPAMIG Experimental Farm in Patrocínio, MG, located in the Alto Paranaíba region, positioned at approximately 18°59′26″ latitude South, 48°58′95″ longitude West, and an altitude of about one thousand meters. The adopted spacing was 3.5 × 1.0 m between rows and between plants, respectively. The soil type is red-yellow latosol, and the topography is flat with a slight slope [27].

Harvesting was conducted in individual plots in 2020 and 2021. The field coffee volume (coffee fruits of mixed maturity) per plot was converted into the number of 60-kg bags of husked coffee produced per hectare (bags ha^−1^) (yield). The experimental design followed randomized blocks with two replicates and ten plants per plot. Plant spacing was 4.0 × 1.0 m. Sowing and crop management were carried out according to the technical recommendations for the species’ cultivation.

### 2.3. Statistical Analyses

#### 2.3.1. Calculation of the MVPi

To assess the phenotypic plasticity of the evaluated genotypes, data obtained from physiological and biochemical assessments were subjected to a multivariate approach using an index that considers multiple evaluated traits. The multivariate plasticity index (MVPi), proposed by Pennacchi et al. [19], was calculated based on the absolute deviation, measured as Euclidean distances in the multidimensional Cartesian plane, between different phenotypic states of the plants, specifically with and without water deficit. The scores for each individual were defined through principal component analysis (PCA).

The MVPi values were calculated at three different stages, corresponding to the same time points as other physiological and biochemical parameters: before the application of water deficit (WW), during its application (WD), and after irrigation resumed (RW). Using the MVPi values for each point, differences in plasticity values were calculated as the slopes of lines (I) between periods: I1 between the initial measurement and the post-imposition of stress points, and I2 between the post-imposition of stress points and the return of irrigation. The I1 and I2 values were then correlated with differences in total biomass accumulation in both above and belowground parts between control and water deficit treatments, as well as with conventional field management productivity. All analyses were performed using R Statistical Software v4.1.2 [28].

#### 2.3.2. Genetic Parameters

Variance estimation and prediction of random effects were conducted using the restricted maximum likelihood/best linear unbiased prediction (REML/BLUP) procedure with the assistance of the SELEGEN-REML/BLUP software, https://www.scielo.br/j/cbab/a/rzZBnWZ4HnvmsvvL9qCPZ5C/?lang=en (accessed on 29 August 2023) [29]. For these analyses, the employed model was: y=Xβ+Zγ+e. Where: $y_{\left(nx1\right)}$ is the vector of phenotypic observations; $\beta_{\left(bx1\right)}$ is the vector of fixed block effects; $\gamma_{\left(gx1\right)}$ is the vector of random genotype effects; $e_{\left(gx1\right)}$ is the vector of errors; $X_{\left(nxb\right)}$ is the block incidence matrix; and $Z_{\left(nxg\right)}$ is the genotype effect incidence matrix.

Using the estimated variance components, individual heritabilities and other coefficients of determination associated with the random effects of the models were calculated as outlined in Resende [29]. The variance components were subjected to a likelihood ratio test at a 5% significance level. Subsequently, the selection index was determined based on the Mulamba and Mock [30] sum of ranks.

## 3. Results

### 3.1. Microclimatic Data in the Greenhouse

The climatic conditions inside the greenhouse during the experimental period are depicted in Appendix A. At the commencement of the experimental period (1 April 2019), the average temperature and relative humidity were 31 °C and 56%, respectively. On the 25th day after the imposition of the water deficit (26 April 2019), when the evaluated genotypes exhibited variability in response to water stress, the average temperature was 29 °C and the relative humidity was 55%. The maximum recorded temperature during the experimental period reached approximately 42 °C. Following the rehydration of the genotypes (20 May 2019), the recorded average temperature and relative humidity were 28 °C and 59%, respectively.

### 3.2. Field Climate Data

The climatic conditions in the field during the experiment were characterized by well-defined dry and rainy seasons. In the year 2019, the average maximum and minimum temperatures were 30.4 °C and 16.1 °C, respectively. The total annual precipitation was 1299 mm (Appendix A). In August 2019, there was an absence of precipitation, with a maximum temperature of 29.6 °C and a minimum of 14.5 °C (Appendix A). In the year 2020, the average maximum and minimum temperatures were 29.4 °C and 15.9 °C, respectively, and the annual precipitation was 1402 mm. In January 2020, there was a precipitation of 282 mm, and the maximum and minimum temperatures were 30.3 °C and 18.7 °C, respectively (Appendix A).

### 3.3. Multivariate Plasticity Index (MVPi) Correlated with Dry Matter and Productivity

The average values of gas exchange and biochemical analyses at three different assessment points are presented in Appendix A. These values were employed for calculating the MVPi.

In the principal component analysis (PCA), a grouping of genotypes at the well-watered (WW) stage is observed at the top-left quadrant in the direction of the arrows, representing photosynthesis (A) and transpiration (E). This indicates that, among all stages, the WW stage exhibited higher photosynthesis and transpiration. Additionally, stomatal conductance (gs) also trends towards the same left side, albeit closer to leaf water status (ψw), at the bottom-left quadrant (Figure 1).

When a water deficit is imposed (WD), the filled squares are positioned closer to the circles, indicating limited metabolic change in well-watered plants despite the change in developmental stages but not in water treatment. In contrast, empty symbols shift towards the top-right quadrant in the opposite direction of A, E, gs, and ψw. This suggests a decrease in A and E due to lower gs, driven by stomatal control in response to reduced water status. Notably, genotypes 8 (Rubi MG 1192) and 5 (MG 311) exhibit higher distances between filled and empty squares, implying greater systemic phenotypic plasticity as proposed by MVPi (Figure 1).

Upon re-irrigation (RW), a new phenotypic arrangement is observed at the bottom of the graph. The distance between filled circles, squares, and triangles of the same color represents phenotypic changes in well-watered plants across the crop cycle. The distances between filled and empty symbols—denoted by lines connecting two identical symbols—represent the average Euclidean distance between well-watered (W) and water-restricted (D) treatments. MVPi defines this distance as the integrated phenotypic plasticity of each genotype (Figure 1).

Figure 2 presents the MVPi of the nine evaluated genotypes at three time points. MVPi serves as an indicator of the integrated phenotypic distance between the control treatment and the water deficit treatment. A higher value indicates a greater phenotypic distance between the irrigated control and water deficit treatments. In the period between initial measurement and post-imposition of stress (I1) on the graph, the phenotypic alteration caused by water deficit imposition is shown in relation to the control.

The inclination between WD and RW points represents the phenotypic alteration caused by re-irrigation after a period of water deficit (I2). A variation in genotype plasticity is observed, with higher plasticity seen in genotypes 8 (Rubi MG 1192), 5 (MG 311), and 1 (MG 270B1), followed by genotypes 6 (MG 279), 3 (MG 364), 7 (MG 308), and 4 (MG 534) exhibiting intermediate values. Conversely, lower plasticity is observed in genotypes 9 (IPR 100) and 2 (MG 270B2). There is a common pattern of reduced phenotypic distance after re-irrigation, indicating similarity between the metabolism of plants subjected to water deficit and those consistently irrigated, except for genotype 9 (IPR 100). Among other genotypes, some show a greater ability to return to metabolic levels similar to the control, such as genotypes 1 (MG 270B1), 2 (MG 270B2), 3 (MG 364), and 8 (Rubi MG 1192).

When evaluating integrated phenotypic plasticity, a correlation was observed between MVPi values and patterns of total biomass loss in the shoot and root parts (Figure 3A–C) between control and water deficit plants. This pattern suggests that genotypes exhibiting greater plasticity in response to water deficits also experienced lower biomass losses compared to irrigated controls.

Likewise, after re-irrigation, a very similar pattern between plasticity and biomass loss is evident. In this case, more negative values indicate higher plasticity, showing that genotypes 8 (Rubi MG 1192), 1 (MG 270B1), and 5 (MG 311), exhibiting higher plasticity, had lower biomass losses at the end of the cycle (Figure 4A–C). Despite genotypes 2 (MG 270B2) and 3 (MG 364) returning to a phenotypic state very close to the control after re-irrigation, there was substantial biomass loss due to the water limitation cycle, indicating reduced plasticity and low adaptation capacity during stress.

When genotypes are compared in relation to field productivity (Figure 5), a consistent pattern emerges wherein those demonstrating higher plasticity in response to water deficit and re-irrigation in greenhouse experiments also exhibit greater field productivity. Based on the I1 of the study, genotypes 8 (Rubi MG 1192), 5 (MG 311), 1 (MG 270 B1), and 6 (MG 279) stood out in terms of plasticity and productivity. In the context of I2, where plasticity reflects the capacity to return to the physiological state after the water deficit period, genotypes 8 (Rubi MG 1192) and 1 (MG 270 B1) excelled.

However, it is important to note that genotype 2 (MG 270 B2) presents an anomaly concerning this linear pattern, as it displays high field productivity despite its low plasticity in the greenhouse.

### 3.4. Genetic Parameters and Genotype Selection

To examine whether the MVPi indices studied at the two inclinations (I1 and I2) could be employed in the selection process, we estimated genetic parameters (Table 2). The likelihood ratio test revealed genetic variability for the indices in both studied inclinations. The average heritability of genotypes indicated the potential for selection with high magnitudes, suggesting that the studied genotypes tend to maintain stable expression of phenotypic plasticity traits (I1 and I2) in response to environmental changes. The coefficients of relative variation showed values of 0.97 and 0.82 for I1 and I2, indicating a high level of variability in characteristics due to genetic and environmental causes. Moreover, the selection accuracy among genotypes demonstrated high precision in inferring genotypic values (0.94 to 0.93), indicating that the experimental setup was appropriate, and the evaluation based on these traits can characterize plastic genotypes.

Inclinations I1 and I2 can be selected as traits for greenhouse selection of progenies with greater plasticity, as these traits exhibit satisfactory genetic parameters (Table 2). According to the Mulamba and Mock index [30] (Table 3), the summed ranks indicated that genotypes 8 (Rubi MG 1192), 5 (MG 311), 1 (MG 270), and 6 (MG 279) were superior in plasticity compared to the other genotypes, corroborating the results presented in Figure 5. In the selection of improved progenies, the predicted genotypic values favored genetic gains relative to the genotypic mean, with an increase of 57.09% in I1 and 36.67% in I2 for the selection of progenies with greater phenotypic plasticity.

## 4. Discussion

Water stress is one of the most significant environmental factors affecting agricultural production, leading to growth reduction, development problems, and decreased yield. As stated simply, lower levels of water in the soil limit its uptake by plants, decreasing plant relative water content and cell turgor. This causes a response of stomata closure, which reduces carbon uptake and biomass accumulation through photosynthesis [31].

Responses to water stress vary among species and within genotypes of the same species, with some being more adapted due to genetic, environmental, and management factors [4,16,17]. Tolerance to stress is determined by these complex elements. Given the increasing demand for drought-tolerant coffee genotypes and the emergence of new automated phenotyping techniques, the need for a more efficient genotypic selection process has significantly intensified [31]. Analyzing these multiple parameters individually and their intercorrelations complicates decision-making in the breeding process. To address this challenge, our research group has been seeking integrated parameters for assessing plasticity that align with breeding selection criteria.

Phenotypic plasticity—the ability of a specific genotype to display multiple phenotypes in response to the environment—can be used to predict plant behavior under stress conditions [32]. This phenomenon has been extensively studied in the context of plant adaptation to adverse environmental conditions [3]. However, there is no consensus regarding the relationship between phenotypic plasticity and plant productivity. This is mainly because phenotypic plasticity under stress conditions is linked to plant survival and yield penalization [33]. This relationship gets even more complicated when phenotypic plasticity is considered a modular response of single traits instead of a complex response with emergent characteristics [34].

Therefore, there is an urgent need for a systemic approach to evaluating phenotypic plasticity, aiming to improve the understanding of the plant system as a whole based on non-reducible characteristics [35,36]. The MVPi is proposed as a phenotypic plasticity index, which comprises multiple traits into a single indicator of plant status—not in a modular way, but as a representation of the whole [16].

In this study, we investigated the relationship between MVPi, biomass accumulation, and the production of genotypes subjected to water limitation. We found that the MVPi fulfilled three main desired characteristics to be considered a potential marker for genetic improvement. Firstly, we observed a clear correlation between MVPi and productivity and productive resilience in different growth environments. Additionally, we noted wide genetic variation among the evaluated genotypes in relation to MVPi, highlighting its relevance as a parameter of interest for selection. Lastly, our results demonstrate that MVPi is heritable, meaning it is genetically transmitted to subsequent generations, solidifying its utility as a valuable tool in agricultural crop improvement programs.

Plants have three main strategies when dealing with stress: avoidance, escape, and tolerance. Avoidance is more related to the mechanisms that try to alleviate the impacts of stress, such as, for instance, stomata closure and leaf rolling; strategies to escape stress are related to shortening the cycle or relocating biomass to root growth; tolerance is linked to the acquired memory of likely mechanisms to maintain plant metabolisms, such as osmotic adjustment and tolerance to desiccation [37]. Based on these three main strategies, there are infinite combinations of metabolic arrangements of single plant parameters, which may be linked to a higher or lower capacity to perform under stress [38]. These arrangements are normally linked to genetic background, which reinforces the need to understand performance under drought as a complex trait for each genotype instead of a trait-by-trait evaluation.

Another important point is the need to better understand the plant shift from production mode to survival mode when environmental conditions are not ideal. Many of the changes in biochemical, physiological, and anatomic parameters linked to survival are also linked to productivity depletion [39]. This dichotomy between survival and production creates a dilemma for the plant and complicates the understanding of which metabolic balance is best for agricultural purposes [40]. Again, we reinforce that the analysis of single traits may be biased indicators of performance and that an integrated evaluation of a plant’s general status may be more conclusive and representative. Of course, such responses do not stay the same throughout the different moments of the drought cycle, such as water limitation and the return of natural or artificial precipitation. For those reasons, it is important to map different tolerance levels during drought and recuperation during rewatering, as we proposed in this study.

When analyzing the I1 of MVPi related to the period of irrigation suspension, we noticed that genotypes 8 (Rubi MG 1192), 5 (MG 311 Hybrid Timor UFV 428-02), 1 (MG 270 B1 Hybrid Timor UFV 377-21), and 6 (MG 279 Hybrid Timor UFV 376-31) show the remarkable ability to adapt to changing water conditions in the greenhouse, which translates into high field productivity. Such adjustments facilitate the preservation of metabolic activities, illustrating an adaptive approach for confronting water stress while sustaining dry matter content and overall productivity. In the context of the second index (I2), which assesses the capacity to return to the physiological state after the water deficit period, genotypes 8 (Rubi MG 1192) and 1 (MG 270 B1) excel. This indicates their exceptional resilience and ability to recover after experiencing water-deficit conditions.

The sensitivity to water deficit of the “Rubi MG 1192” cultivar (genotype 8), as described by other authors [17], was not observed in this study. The observed tolerance behavior may be related to greater phenotypic plasticity, as indicated by the larger Euclidean distance between the control and water deficit treatments (Figure 1, G8, distance between filled and empty squares). It is evident that the control treatment in the WD stage has a phenotypic state similar to the same treatment in the previous stage (WW), shifted in the direction of the vectors A, E, gs, and ψw. On the other hand, the plants under water deficit are shifted to the far-right side of the Cartesian plane, indicating stomatal closure and increased water-use efficiency. The tolerance mechanism may be related to increased carbon fixation during periods of higher water availability and enhanced control of water loss during periods of water deficit.

Among the most common plastic genotypes, accessions 5 (MG 311 Hybrid Timor UFV 428-02), 1 (MG 270 B1 Hybrid Timor UFV 377-21), and 6 (MG 279 Hybrid Timor UFV 376-31) belong to the Hybrid Timor germplasm (Table 1), commonly used in breeding programs as a source of resistance to major coffee diseases [41]. In another study, these genotypes were found to have leaf anatomical structures that optimize water transport and prevent excessive transpiration, as well as physiological mechanisms that minimize impacts during dry periods [17]. This adaptive capacity is crucial in future scenarios of climate change, where extreme events such as prolonged droughts may become more frequent.

In plants subjected to recurring water stress, such as coffee, the ability to restore their pre-stress behavior is crucial for resuming growth and productivity, as demonstrated by Beacham et al. [42]. Both recovery ability and drought tolerance are essential, as indicated by Hassan et al. [3]. This phenomenon was observed in genotypes 1 (MG 270 B1 Hybrid Timor UFV 377-21), 2 (MG 270 B2 Hybrid Timor UFV 377-21), 3 (MG 364 Hybrid Timor UFV 442-42), and control 8 (Rubi MG 1192), which exhibited a greater ability to return to metabolic levels similar to the irrigated control.

Phenotypic plasticity ranges from stable to highly plastic in genotypes [43]. The “IPR 100” cultivar demonstrated lower plasticity, possibly related to different drought tolerance mechanisms [20]. Its distinction may stem from hybridization origin, “Catuaí” × coffee (“Catuaí” × coffee genotype from the “BA-10” series) bearing *C. liberica* genes, diverging from other materials in the Hybrid Timor germplasm (Table 2). Future studies can explore these mechanisms for sustainable agronomic approaches in the face of climate change.

The concept of physiological breeding [44] proposes the use of parameters related to plant metabolism—primarily those of physiological and biochemical nature—as indicators for selecting genotypes in plant breeding. The growing capacity to measure these parameters, driven by high-throughput phenotyping methods, is remarkable. However, this progress demands the development of new statistical approaches that translate phenotyping advancements into substantial genetic gains [45]. In this study, genetic parameters were estimated, indicating promising prospects for selecting individuals with greater phenotypic plasticity. Subsequently, the application of the Mulamba and Mock selection index [30] was proposed, using the plasticity index during the interval between the initial measurement and the subsequent point after stress imposition (I1), as well as during the period between post-stress imposition and irrigation return (I2), when the plants were under water stress conditions. By amalgamating the genotype rankings evaluated by Mulamba and Mock [30], we selected genotypes 8 (Rubi MG 1192), 5 (MG 311 Hybrid Timor UFV 428-02), 1 (MG 270 B1 Hybrid Timor UFV 377-21), and 6 (MG 279 Hybrid Timor UFV 376-31). These selections exhibited significant enhancements in phenotypic plasticity, with increments of 57.09% in I1 and 36.67% in I2. This reaffirms the consistency of prior findings across diverse assessments (Figure 1, Figure 2 and Figure 3).

The outcomes of this investigation align cohesively with antecedent studies, which also accentuated the pivotal role of phenotypic plasticity as a valuable tool in prognosticating plant responses in the age of climate change [23,43]. The prognostication of genotypic responses to diverse environmental stress scenarios stands as a pivotal imperative in the realm of devising strategies for the management and selection of more resilient and productive cultivars.

Our research demonstrates a clear advantage for well-watered plants, which consistently exhibit higher rates of photosynthesis and transpiration. Moreover, genotypes displaying greater plasticity in response to water deficits also showed reduced biomass losses compared to their irrigated counterparts. This decline in photosynthesis (A) and transpiration (E) under water-deficit conditions is primarily attributed to reduced stomatal conductance. Genotypes 8 and 5 stand out for their impressive ability to adapt to changing environmental conditions, indicating a higher level of phenotypic plasticity as assessed by the MVPi metric. This highlights the significance of this adaptive mechanism in enhancing plant productivity in water-stress environments.

Our findings emphasize that phenotypic plasticity plays a pivotal role in plant responses to water stress. Genotypes with a higher degree of plasticity, which are capable of meaningfully adjusting their physiological and biochemical parameters in response to environmental cues, experienced reduced biomass losses in both aerial and root components. This positive effect ultimately contributes to an overall increase in productivity. In a related study, Bhusal et al. [5] observed a similar trend in apple plants subjected to drought conditions. They noted a shift in resource allocation towards structural investment at the expense of photosynthesis upregulation. The examined apple cultivar exhibited a decrease in key photosynthetic parameters in response to a water deficit, accompanied by a significant increase in leaf mass per unit area. This underscores the importance of resource allocation strategies in plant adaptation to water scarcity.

The MVPi emerged as a valuable instrument for assessing the adaptive capacity of genotypes and forecasting their performance across varying climatic scenarios. Heightened plasticity was discernible within the GC-MG accessions: MG 311 Hybrid Timor UFV 428-02, MG 270 Hybrid Timor UFV 377-21, and MG 279 Hybrid Timor UFV 376-31, as well as the Rubi MG 1192 cultivar. It is imperative to underscore that while phenotypic plasticity has demonstrated benefits in the context of biomass preservation and productivity under water stress, substantial uncharted realms remain within this sphere of inquiry. Prospective studies could delve into the molecular underpinnings of phenotypic plasticity and the adaptive response of genotypes to an array of environmental stimuli. A comprehensive understanding of these processes is paramount to formulating sustainable agronomic paradigms in the face of global climatic transformations.

## 5. Conclusions

The adaptive capacity to water stress varies among genotypes, with notable examples being MG 311—Hybrid Timor UFV 428-02, MG 270—Hybrid Timor UFV 377-21, MG 279—Hybrid Timor UFV 376-31, and the Rubi MG 1192 cultivar.

The multivariate phenotypic plasticity index (MVPi) proved to be a valuable tool for assessing and predicting the performance of genotypes under different climatic conditions.

## Figures and Tables

**Figure 1 plants-12-04029-f001:**
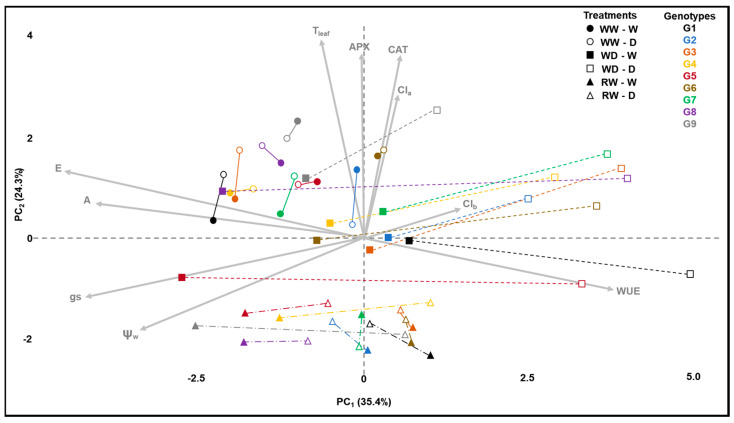
Principal Component Biplot (1st and 2nd PCs) with the average points for all the treatments. Circle: stage before the imposition of water deficit treatment (WW); square: stage after the imposition of the water deficit (WD); triangle: stage after the rewatering (RW). W—well-watered treatment (control, full symbols); D—drought treatment (empty symbols). The lines connecting points represent the MVPi as the Euclidean distance.

**Figure 2 plants-12-04029-f002:**
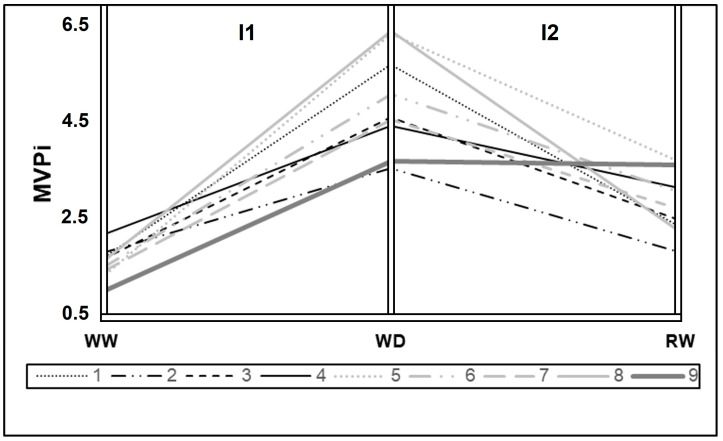
Plasticity of *Coffea arabica* L. genotypes subjected to water deficit in WW (pre-treatment period), WD (25 days after irrigation cutoff), and RW (re-irrigation after water deficit period). Genotypes: 1: MG 270, 2: MG 270, 3: MG 364, 4: MG 534, 5: MG 311, 6: MG 279, 7: MG 308, 8: Rubi MG 1192, and 9: IPR 100. I1—period between initial measurement and post-imposition of stress point; I2—period between post-imposition of stress point and irrigation recovery.

**Figure 3 plants-12-04029-f003:**
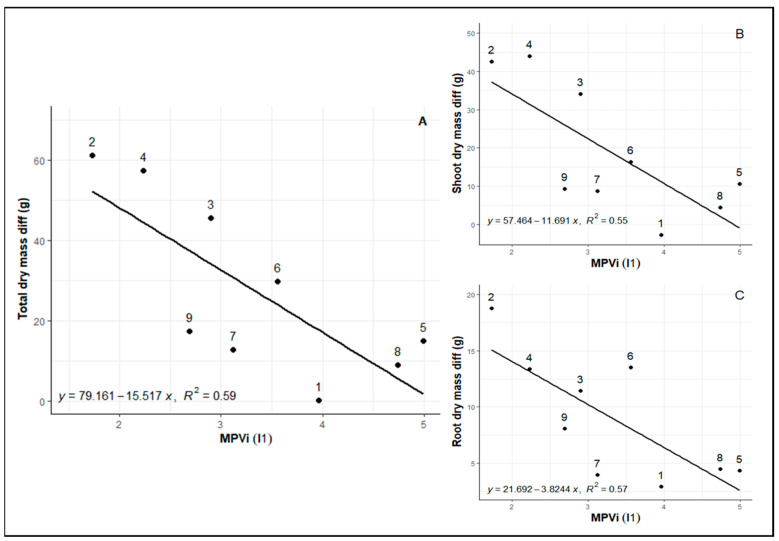
(**A**) Relationship between total dry mass, (**B**) above-ground dry mass, and (**C**) root dry mass with changes in MVPi between WW (pre-treatment period) and WD (25 days after irrigation cutoff), represented by the slope I1 (period between initial measurement and post-imposition of stress point).

**Figure 4 plants-12-04029-f004:**
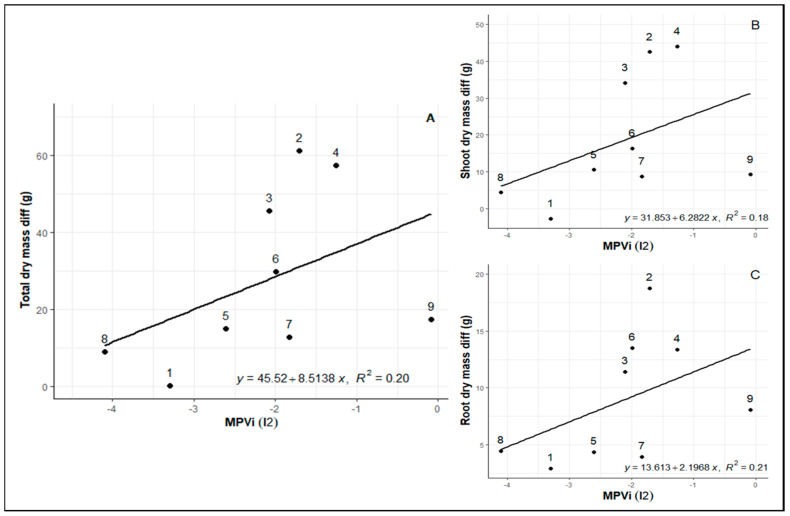
(**A**) The relationship between total dry mass, (**B**) above-ground dry mass, and (**C**) root dry mass with changes in MVPi between WD (25 days after irrigation cutoff) and RW (re-irrigation after water deficit period). Represented by the slope I2 (period between post-imposition of stress point and return of irrigation).

**Figure 5 plants-12-04029-f005:**
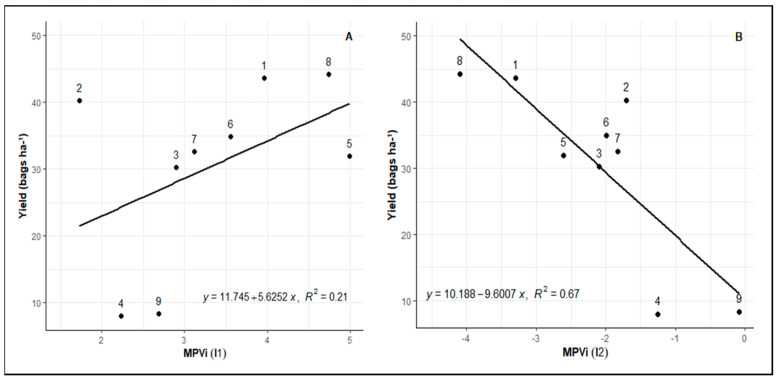
Relationship between MVPi and genotype field productivity. (**A**) The relationship between field productivity and with changes in MVPi between WW (pre-treatment period) and WD (25 days after irrigation cutoff), represented by the slope I1 (period between initial measurement and post-imposition of stress point) obtained in greenhouse experiments. (**B**) The relationship between field productivity and with changes in MVPi between WD (25 days after irrigation cutoff) and RW (re-irrigation after water deficit period). Represented by the slope I2 (period between post-imposition of stress point and return of irrigation) obtained in greenhouse experiments.

**Table 1 plants-12-04029-t001:** Identification and genealogy of genotypes from the Epamig Germplasm Collection (GC-MG).

Number	Identification	Genealogy
1	MG 270 ^1^	Hybrid Timor UFV 377-21
2	MG 270 ^2^	Hybrid Timor UFV 377-21
3	MG 364	Hybrid Timor UFV 442-42
4	MG 534	BE 5 Wush-Wush × Hybrid Timor UFV 366-08
5	MG 311	Hybrid Timor UFV 428-02
6	MG 279	Hybrid Timor UFV 376-31
7	MG 308	Hybrid Timor UFV 427-55
8	Rubi MG1192	Catuaí × Mundo Novo
9	IPR 100	“Catuaí” × Coffee Plant (“Catuaí” × Coffee genotype from the ‘BA-10’ series) carrying genes from *C. liberica.*

^1^ MG 270 block 1; ^2^ Selection of plants (1, 3, and 6) from the MG 270 accession in block 2.

**Table 2 plants-12-04029-t002:** Estimates of genetic parameters and predicted additive genetic values for the inclinations between the initial measurement and the post-imposition of the stress point (I1) and between the post-imposition of the stress point and the return of irrigation (I2).

	$\sigma^{2}$	$h_{mg}^{2}$	$AC_{gen}$	$CV_{r}$	Mean
I1	1.07 *	0.90	0.94	0.97	3.31
I2	1.15 *	0.86	0.93	0.82	−2.10

$\sigma^{2}$: genotypic variance; $h_{mg}^{2}:$ heritability of genotype average, assuming complete survival; $AC_{gen}:$ selective accuracy; $CV_{r}:$ relative coefficient of variation; mean: the overall mean of the experiment; I1: period between initial measurement and post-imposition of the stress point; I2: period between post-imposition of the stress point and return of irrigation. * Significance is determined by the likelihood ratio test at 5% probability.

**Table 3 plants-12-04029-t003:** Predicted additive genetic values for MVPi inclinations I1 and I2 and estimated selection gain (SG%) of the top four (*) progenies with higher phenotypic plasticity.

Number	Genotype	I1	I2	Ij
8	Rubi MG1192	4.59	−3.82	3 *
5	MG 311	4.81	−2.53	4 *
1	MG 270 ¹	3.89	−3.13	5 *
6	MG 279	3.53	−2.00	9 *
3	MG 364	2.94	−2.08	10
7	MG 308	3.13	−1.87	11
9	IPR 100	2.75	−0.36	16
4	MG 534	2.34	−1.37	16
2	MG 270 ²	1.89	−1.76	16
*M_sm_*		4.20	−2.87	5.25
*SG*%		57.09	36.67	

*M_sm_*: selected progenies mean. ^1^ MG 270 block 1; ^2^ Selection of plants (1, 3, and 6) from the MG 270 accession in block 2.

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
