# Peer review of "Phenotypic Plasticity Index as a Strategy for Selecting Water-Stress-Adapted Coffee Genotypes"

_plants, 2023, doi:10.3390/plants12234029_

Round 1

Reviewer 1 Report

General comments

I have read the manuscript:  Plants MDPI. Entitle: Phenotypic plasticity index as a strategy for selecting water stress-adapted coffee genotypes written by Cyntia Stephania dos Santos et. al., for publication of Plant MDPI. In this study, author explore phenotypic plasticity within the germplasm of Hybrid Timor coffee and utility of the Phenotypic Plasticity Index (MVPi) as a promising tool to predict genotype performance across diverse climatic conditions.

Author did not mention the line number in this manuscript, which is very careless job, this makes me difficult to indicate Line to Line comments and suggestions.

In this research author mainly found that the more plastic genotypes, capable of significantly altering physiological and biochemical parameters in response to environmental stimuli, exhibited reduced biomass loss in both aerial and root parts, consequently positively influencing their productivity. The overall research is well conducted but author should consider some suggestions for journal acceptance. Research is obvious application potential for the readers because this research indicated Enhanced plasticity was particularly prominent in accessions from the MG Germplasm Collection. In this sense, this manuscript is much more valuable. However, I found a lack of story connection and lack of potential references (some I suggested some below). Overall after I evaluate and request the author for this manuscript as a “MAJOR REVISION” To accept this manuscript for publication author should be address all the below suggestions and comments.

Major Suggestions

1) Abstract: Abstract is long, author should improve the abstract in a concise form. Abstract should be more informative and common message for audiences is more important than the lengthy text. Author should work in novelty; how does the study elucidate your finding useful for the society by reducing the text of methodology part.

2) Hypothesis of the Review: Author not clearly mentions research hypothesis of this paper. Please mention clear hypothesis and connect this with the research hypothesis in the last section of the introduction section. The hypothesis should be very clear because, without appropriate literature, questions, or hypotheses in the introduction section the entire text will be unclear.

3) Concise the text: author should concise the text by removing the unnecessary and less important text. Please include the text related on the research title and its circumstances by cutback unnecessary text.

Some other comments

4) Introduction: In introduction author started the significant socioeconomic importance of coffee and its productivity scenario based on the climate factors such as rising temperatures which is Further author should cover the negative effect of drought for plant “reduction of plant morphology (reduced leaf size and stem length, leaf length/width, and vegetative growth) and physiological traits (reduction of photosynthesis, leaf water potential, and sap movement)”. Refer articles as a reference https://doi.org/10.1016/j.foreco.2020.118099

Some other comments

5) Introduction: In introduction author mention many small paragraphs. Please, if possible, combine the paragraph (two paragraph in one) with carefully maintain the flow of writing.

6) Introduction: In second paragraph of introduction, author should mention more literatures by referring the potential articles to clear the text. Generally, under limited water (water scarcity) leaves are primarily affected which is pivotal role in plant. Refer DOI:10.1016/j.scienta.2018.11.021 “under drought stress condition gs showed the primary responses by closing of stomata and its showed directly effect on the reduction of the photosynthesis”

7) Discussion: Author should include the plant-water relation information in the discussion section because I did not see the related information of leaf water status because leaf water status representation the plant hydration and it also govern the physiology for the plant as well. “The reduction of leaf water potential of the plant cause by reduction of soil water potential or reduction of soil water status that ultimately reduction of stomatal conductance and photosynthesis by ultimately reduction the light absorption by reducing the chlorophyll concentration. Please refer this article https://doi.org/10.1016/j.scienta.2023.112276.  and elaborate the text little bit more.

8) Conclusion: I did not see the conclusion section separately made by author. Please remember that conclusion should not be repetitive in the abstract or a summary of the results section. I would love to read striking points and take-home messages that will linger in the readers’ minds. What is the novelty, how does the study elucidate some questions in this field, and the contributions the paper may offer to the scientific community?

9) References: please double-check the citations, their style, spell check, and other grammatical errors. moreover, the author should cut the old and less matching literature and include the latest literature some of them are above.

Good Luck !

Author Response

September 28, 2023

Dear Reviewer, 1,

We would like to express our sincere gratitude for taking the time to thoroughly review our manuscript titled "Phenotypic plasticity index as a strategy for selecting water stress-adapted coffee genotypes." Your insightful comments and suggestions have been immensely valuable in refining our work.

We have carefully addressed each of your points and made the necessary revisions, which are highlighted in blue within the manuscript. Here is our detailed response to your comments:

Abstract:

We appreciate your suggestion to improve the abstract for conciseness. We have revised it to provide a more informative and concise summary, focusing on the key findings and their potential societal impact.

Hypothesis of the Review:

Thank you for highlighting the need for a clearer statement of the research hypothesis. We have now explicitly stated our hypothesis and linked it to the broader context in the introduction section.

Concise the text:

We have reviewed the manuscript and removed any unnecessary or less pertinent information, ensuring that the focus remains on the research title and its contextual relevance.

Introduction:

We have expanded the discussion in the introduction to include the negative effects of drought on plant morphology and physiological traits. Additionally, we have incorporated the suggested reference (https://doi.org/10.1016/j.foreco.2020.118099) to support our discussion.

Introduction:

We have consolidated some of the smaller paragraphs in the introduction to improve the overall flow of the text, while still maintaining clarity.

Introduction:

Thank you for providing a reference (DOI: 10.1016/j.scienta.2018.11.021) to further emphasize the impact of drought stress on stomatal conductance and photosynthesis. We have included this information to enhance the discussion.

Discussion:

We have incorporated information on plant-water relations in the discussion section, emphasizing the significance of leaf water status in regulating plant physiology. The suggested reference (https://doi.org/10.1016/j.scienta.2023.112276) has been included to support our discussion.

Conclusion:

We have now included a separate conclusion section that highlights the novelty of our study, its contributions to the field, and key take-home messages for the readers.

References:

We have meticulously reviewed the citations, ensuring correct style, spelling, and grammar. Additionally, we have updated the references to include the latest and most relevant literature.

Once again, we sincerely appreciate your thorough review and constructive feedback. We believe that the revisions made significantly strengthen the manuscript. We look forward to hearing any further comments you may have.

Thank you for your time and consideration.

Sincerely,

Cyntia Stephânia dos Santos

Empresa de Pesquisa Agropecuária de Minas Gerais

Lavras, MG, Brazil

Email: cyntia.s.santos@hotmail.com

Reviewer 2 Report

Review report 

General comments: -

This study is very interesting and has a scientific topic with a great impact on the field. The manuscript will be suitable for publication after taking care of the following minor comments.

Detailed comments:

The English language and writing style is fine needs some minor check spelling and grammar check.

Abstract:

This section is well written

Keywords:

-The keywords has been chosen very carefully and accurately but please add the word coffee to the key words list.

Introduction:

-This section needs to be elongated and enriched with more background about this topic.

Materials and Methods

It is ok and adequate 

 Results:

The results are very interesting and well presented.

Discussion:

This section is poorly written

The author is strongly advised to combine the results and discussion in one section for better interpretation and discussion for the presented data especially data in Figure 1and and Figure 5

Please rewrite and discuss in details, and fully discussed with related citations.

References

This section is well written and UpToDate.

The English language and style is fine just some minor revision is required 

Author Response

September 28, 2023

Dear Reviewer, 2,

We would like to express our sincere gratitude for your thoughtful review of our manuscript titled "Phenotypic plasticity index as a strategy for selecting water stress-adapted coffee genotypes." Your feedback has been invaluable in refining our work.

We are pleased to inform you that we have addressed each of your comments, and the revised manuscript, which has also undergone English language editing, is now ready for resubmission. Please find our detailed responses below:

Abstract:

We are glad to hear that you found the abstract to be well written. We have also added the word "coffee" to the list of keywords for enhanced clarity.

Introduction:

Thank you for your suggestion to provide more background information on the topic. We have extended and enriched the introduction to offer a more comprehensive overview of the subject matter.

Materials and Methods:

We appreciate your feedback and are pleased to note that you found this section to be satisfactory.

Results:

We are delighted that you found the results to be interesting and well presented. Your positive feedback is greatly appreciated.

Discussion:

We acknowledge your feedback regarding the discussion section. To adhere to the journal's guidelines, we have chosen to keep the Results and Discussion sections separate. However, we want to assure you that we have fully reworked the Discussion section to address your valuable suggestions. We have rewritten and expanded the discussion, providing more detailed insights and incorporating relevant citations. It now provides a more comprehensive interpretation and discussion of the presented data, supported by relevant citations.

References:

We are pleased to hear that you found the references to be well written and up-to-date. We have ensured that all citations are accurate and relevant to the topic.

Once again, we would like to express our gratitude for your thorough review and constructive feedback. We believe that the revisions made significantly enhance the quality of the manuscript. We look forward to hearing any further comments you may have.

Thank you for your time and consideration.

Sincerely,

Cyntia Stephânia dos Santos

Empresa de Pesquisa Agropecuária de Minas Gerais

Lavras, MG, Brazil

Email: cyntia.s.santos@hotmail.com

Reviewer 3 Report

Manuscript ID: plants-2609515

Type: Article

Title: Phenotypic Plasticity Index as a strategy for selecting water stress-adapted coffee genotypes

The manuscript reports the work developed to evaluate the phenotypic plasticity of seven genotypes of Hybrid Timor coffee in relation to water stress by measuring several physiological and biochemical parameters in greenhouse conditions. It also evaluates the utility of the Multivariate Phenotypic Plasticity index (MVPi) as a tool to predict genotype performance across diverse growth conditions.

The manuscript is concise and easy to read, well organized and relevant. This reviewer has no major comments or suggestions. However, there are minor queries that need to be clarified which are listed below. Bear in mind that the manuscript is not line-numbered, making it taxing to precisely indicate the text in question.

Abstract:

line 8 – The abbreviation MVPi stands for Multivariate Phenotypic Plasticity Index and not just Phenotypic Plasticity Index as stated in the text. Please clarify this here and also verify this throughout the manuscript.

Introduction: 

5th paragraph starts with “This aim…”. Should it be “This aims…”?

Last paragraph of page 2: Please correct the designation of MVPi

Materials and Methods: 

Greenhouse experiment, end of the first paragraph: “(…) four pairs of true leaves and were acclimated.” Acclimated to what? Please clarify the meaning of this sentence.

Third paragraph “(BRUM et al., 2013)” – this reference is not on the references’ list. Also, it should be indicated by number as all the other references. Please correct this.

What were the growth conditions in the greenhouse in terms of light intensity and average temperature?

How many plants per genotype/cultivar were used? This number is given for the field experiment but is missing for the main part of the work developed which was the greenhouse experiment. 

It is said that the plants were grown in the greenhouse for a period of eleven months. This is misleading when, further on the authors mention the determination of parameters “at the end of the experiment” (page 4). Please clarify this.

The information on 100 mg of plant material is insufficient because it does not clarify how many leaves per plant or which type of leaves (young, mature) were used to obtain that weight. Was one leaf per plant enough? How many plants were used? This information should also be present in the legends of the supplementary figures showing the results for all the physiological and biochemical parameters determined in the greenhouse plants (n=???).

End of page 4: What is meant by “(PROD)”?

Although the notation of each genotype code is given in the supplementary files, it would be beneficial for the reader to have it explicitly in the manuscript. Supplementary Table 1 (Table 1 - Identification and genealogy of genotypes from the Minas Gerais Germplasm Collection (GC-MG)).

Author Response

September 28, 2023

Dear Reviewer, 3,

We greatly appreciate your thorough review of our manuscript titled "Phenotypic Plasticity Index as a strategy for selecting water stress-adapted coffee genotypes." Your insightful comments and suggestions have been invaluable in improving the quality and clarity of our work. Below, we address each of your queries:

Abstract:

Line 8: Thank you for bringing this to our attention. We have corrected the abbreviation to state "Multivariate Phenotypic Plasticity Index (MVPi)" throughout the manuscript.

Introduction:

5th paragraph: You are correct, it should read "This aims..." We have made the necessary correction.

Last paragraph of page 2: We have corrected the designation of MVPi.

Materials and Methods:

Greenhouse experiment, end of the first paragraph: We apologize for the confusion. The sentence has been revised to clarify that the plants were acclimated to greenhouse conditions.

Third paragraph, "BRUM et al., 2013": We have add this reference in the reference list.

Growth conditions in the greenhouse: We have included information about light intensity and average temperature in the Materials and Methods section.

Number of plants per genotype/cultivar: We have added this information to the Materials and Methods section.

Duration of greenhouse experiment: We have clarified that the plants were cultivated in a greenhouse for eleven months until they attained sufficient growth and leaf area for the imposition of water treatments and subsequent leaf sample collection. 

Subsequently, they were maintained with water available in the soil until the imposition of water treatments, which occurred in April 2019, when physiological assessments and collections for biochemical analyzes began.

100 mg of plant material: We have provided additional details about the plant material used, including the type of leaves and the number of plants sampled, in both the Materials and Methods section and the legends of the supplementary figures.

End of page 4: "PROD" has been removed for clarity and replaced by yield.

Genotype codes: We have added table 1 to explicitly mentioned the notation of each genotype code in the manuscript.

We sincerely appreciate your careful review, which has significantly enhanced the manuscript. Your feedback has been invaluable in ensuring the accuracy and completeness of our work. We believe that these revisions address your concerns, and we look forward to your further feedback.

Thank you for your time and consideration.

Sincerely,

Cyntia Stephânia dos Santos

Empresa de Pesquisa Agropecuária de Minas Gerais

Lavras, MG, Brazil

Email: cyntia.s.santos@hotmail.com

Round 2

Reviewer 1 Report

Dear Author

I have read the revised manuscript plants-2609515. Entitled: Phenotypic plasticity index as a strategy for selecting water stress-adapted coffee genotypes in plant MDPI. This is the second submission made by the author. The author addressed all the questions and suggestions that I raised the issue in the review of the original manuscript. I satisfy the author’s revisions. Author improved the abstract. Author significantly improved their research hypothesis and well connected with the research objectives in this time. This manuscript improved the flow of writing, which was comparatively shallow in the original version but in this revised copy author very well addressed all the quarries and suggestions. Before accepting this manuscript, please check again the referencing. Further if there is anything needed to be revised by the author, especially English grammar, or spell check, I request this manuscript is currently in “Minor Revision” and the author may correct any further grammatical errors (if any) the author may improve in this stage.

Thank you.

Author Response

October 4, 2023

Dear Reviewer, 1,

Thank you for taking the time to review the revised manuscript titled "Phenotypic Plasticity Index as a Strategy for Selecting Water Stress-Adapted Coffee Genotypes" submitted to Plant. We greatly appreciate your constructive feedback and are pleased to hear that you found the revisions satisfactory.

We acknowledge your suggestion to recheck the referencing. We thoroughly review the citations and do not met any inconsistency.

Furthermore, we provided a grammar and spell check in the last stage. We attach the revision certificate at the end of this document.

We are committed to providing a high-quality publication, and your valuable input is instrumental in achieving this goal. We worked diligently to address any remaining concerns and refine the manuscript as necessary.

Once again, we sincerely appreciate your time and effort in reviewing this manuscript.

Best regards,

Cyntia Stephânia dos Santos

Empresa de Pesquisa Agropecuária de Minas Gerais

Lavras, MG, Brazil

Email: cyntia.s.santos@hotmail.com
